# New Analytical Approach to Quinolizidine Alkaloids and Their Assumed Biosynthesis Pathways in Lupin Seeds

**DOI:** 10.3390/toxins16030163

**Published:** 2024-03-21

**Authors:** Dvory Namdar, Patrick P. J. Mulder, Eyal Ben-Simchon, Yael Hacham, Loai Basheer, Ofer Cohen, Marcelo Sternberg, Oren Shelef

**Affiliations:** 1Natural Resources, Institute of Plant Sciences, Agricultural Research Organization, 68 Maccabim Road, P.O. Box 151590, Rishon Le Tzion 7505101, Israel; dvoran@volcani.agri.gov.il (D.N.); eyal.ben-simchon@mail.huji.ac.il (E.B.-S.); 2Wageningen Food Safety Research (WFSR), Wageningen University & Research, Akkermaalsbos 2, 6708 WB Wageningen, The Netherlands; patrick.mulder@wur.nl; 3The R.H. Smith Institute of Plant Science and Genetics in Agriculture, Faculty of Agriculture, Food and Environment, The Hebrew University of Jerusalem, Rehovot 7610001, Israel; 4Laboratory of Plant Science, Migal—Galilee Technology Center, Kiryat Shmona 1210001, Israel; yaelh@migal.org.il; 5Food Sciences Department, Faculty of Sciences and Technology, Tel Hai College, Upper Galilee 1220800, Israel; loaibasheer@telhai.ac.il; 6School of Plant Sciences and Food Security, Faculty of Life Sciences, Tel Aviv University, Tel Aviv 6997801, Israel; ofercoh77@gmail.com (O.C.); marcelos@tauex.tau.ac.il (M.S.)

**Keywords:** *Lupinus pilosus*, *Lupinus palaestinus*, quinolizidine alkaloids, LC–MS/MS, GC–MS, toxicity, biosynthesis

## Abstract

Alkaloids play an essential role in protecting plants against herbivores. Humans can also benefit from the pharmacological effects of these compounds. Plants produce an immense variety of structurally different alkaloids, including quinolizidine alkaloids, a group of bi-, tri-, and tetracyclic compounds produced by *Lupinus* species. Various lupin species produce different alkaloid profiles. To study the composition of quinolizidine alkaloids in lupin seeds, we collected 31 populations of two wild species native to Israel, *L. pilosus* and *L. palaestinus*, and analyzed their quinolizidine alkaloid contents. Our goal was to study the alkaloid profiles of these two wild species to better understand the challenges and prospective uses of wild lupins. We compared their profiles with those of other commercial and wild lupin species. To this end, a straightforward method for extracting alkaloids from seeds and determining the quinolizidine alkaloid profile by LC–MS/MS was developed and validated in-house. For the quantification of quinolizidine alkaloids, 15 analytical reference standards were used. We used GC–MS to verify and cross-reference the identity of certain alkaloids for which no analytical standards were available. The results enabled further exploration of quinolizidine alkaloid biosynthesis. We reviewed and re-analyzed the suggested quinolizidine alkaloid biosynthesis pathway, including the relationship between the amino acid precursor l-lysine and the different quinolizidine alkaloids occurring in seeds of lupin species. Revealing alkaloid compositions and highlighting some aspects of their formation pathway are important steps in evaluating the use of wild lupins as a novel legume crop.

## 1. Introduction

Over millions of years, the long evolutionary tradeoff between plants and the animals that feed on them has led to the production of numerous phyto-compounds that prevent plants from being over-eaten [1]. A prominent and evolutionarily stable strategy in plants is the production of toxic, repellent, or anti-nutritive metabolites [2]. These substances protect plants against herbivores, limit plant feeding, and decrease plant biomass loss [3]. Many such compounds are toxic to herbivores and vertebrates, including humans. They are also harmful to other microorganisms, such as viruses, bacteria, and fungi, and thus, could be natural biocides to help combat pathogens and treat diseases, including cancers [3].

Quinolizidine alkaloids (QAs) are a group of these toxic compounds. QAs are characterized by their quinolizidine ring structure [4] and are produced as secondary metabolites by members of the genus *Lupinus* [5,6]. QAs offer plants efficient protection from insect pests, and their accumulation in significant quantities complicates lupin consumption [7]. The EU has not regulated a QA human/mammal consumption threshold. However, the following QAs will likely be of interest for future legislation: albine, anagyrine, angustifoline, multiflorine, sparteine, lupanine, isolupanine, and 13-hydroxylupanine. Except for anagyrine, all of these are QAs typically found in white and blue lupins (*L. albus* and *L. angustifolius*, respectively), the two main species that are consumed in the EU. To date, only Australia and New Zealand have regulated the allowable QA threshold level in lupins, limiting it to 200 mg per kg of lupin product [8].

Despite their potential uses for medicine and plant protection, information on the toxicity of QAs occurring in *Lupinus* species is limited [4,9], and so far, no comprehensive human health risk has been established [4,10,11,12,13]. Of the various QAs found in edible lupins, sparteine is generally regarded as the most poisonous, although its association with the intake of industrially produced lupin foods has not been reported [4,14].

Four lupin species have been domesticated and are commercially grown as legume crops: *L. angustifolius* (blue or narrow-leaved lupin), *L. luteus* (yellow lupin), *L. albus* (white lupin), and *L. mutabilis* (pearl or Andean lupin), of which, *L. angustifolius* is the most produced [7]. The chemical structures of many QAs are well characterized [15], and lupin cell cultures and enzyme assays have been employed to identify some biosynthetic enzymes and pathway intermediates [7]. However, a complete understanding of the QA biosynthetic pathway remains unresolved. QA biosynthesis is light regulated, being stimulated during the day, and occurs in the stroma of leaf chloroplasts [16]. Once produced, QAs are phloem transported and stored in vacuoles of plant organs [17]. Stems and leaves accumulate alkaloids, but the alkaloids disappear during leaf senescence. Lupin seeds, on the other hand, accumulate QAs as they mature.

We wish to explore the ability to use wild lupin species as a means to enhance diversity and resilience in local crops [18] or as a supplementary protein crop [19]. However, lupin bitterness and health risks are mainly attributed to their alkaloid content and QA profile ([4] and references therein). To cultivate wild lupins into a food crop, a thorough evaluation, and ultimately, control over the alkaloid concentrations and profile, must be achieved. To control legumes’ bitterness, people have often used debittering techniques such as fermentation, soaking, or salt-rich cooking, or they have focused on breeding sweeter seeds [7]. Generally, breeding for sweeter lupins usually aims to relocate QA production and accumulation to lupin vegetative tissues to preserve insect resistance, while reducing its production in seeds to keep the QA levels within industry demands [20,21]. Another strategy is to screen for wild lupin populations naturally low in QA levels and to grow them in suitable agricultural conditions. A third approach is to manipulate the QA biosynthesis pathway by blocking the biosynthesis pathway of non-desired alkaloids or of their precursors. In all of these scenarios, obtaining knowledge regarding the QA profile, particularly its biosynthesis, is crucial for transforming wild lupins into an industrial crop and food ingredient.

The QAs present in the abovementioned species may be divided into four major structural groups: lupanine (including angustifoline), lupinine, multiflorine, and sparteine [4]. A fifth, structurally independent group has an alpha-pyridone structure, including thermopsine, cytisine, and anagyrine, and is considered highly toxic. The structural diversity of QAs is largely species dependent, and each lupin species has a unique alkaloid profile [15,17,22]. This quinolizidine-based chemo-diversity is considerable. For example, 46 alkaloids were identified in 22 cultivars derived from three different Italian species grown in Sicily, including some alkaloids not shared with other edible cultivars [22]. Quantitative analyses demonstrated large differences in the total QA content of lupin species, ranging from 0.003 to 32.8 mg/g in the fresh leaves of eight species [17]. Many questions, such as edibility (i.e., bitterness and toxicity) and pharmaceutical potential, are strongly related to what governs the vast diversity of both QA content and relative concentration, namely the alkaloid biosynthesis mechanism. In a recent DNA molecular study, Bermúdez-Torres and colleagues [23] showed that QA diversity development is affected by geographical and evolutionary origin [23]. Nevertheless, the molecular mechanisms underlying the biosynthesis of lysine-derived alkaloids, including QAs, are not yet fully understood [16]. Advances in the understanding of lysine-derived alkaloids led researchers to suggest a theoretical biosynthesis pathway [7,13,16]. According to this hypothesis, lysine decarboxylase catalysis is the first step in the biosynthetic pathway of QAs [16]. This model also refines the molecular role of lysine decarboxylase activity in plants, highlighting lysine decarboxylase as the enzyme catalyzing the first initiated step of QA biosynthesis and depicting the likely relationships between the different types of QAs and derivatives [16].

Most studies on lupin-derived QAs focus on the four cultivated species. In this study, we evaluated the QA composition of two wild species native to Israel—*L. pilosus* and *L. palaestinus*. *L. pilosus*, common in Mediterranean shrublands, which have large seeds and a high tolerance to various soil conditions, while *L. palaestinus* prospers on sandy calcareous soils. Only limited information on the QA content of *L. pilosus* and *L. palaestinus* is available in the literature [22,24]. Among other *Lupinus* species, Święcicki et al. [25] studied the QA content of *L. pilosus* and *L. palaestinus* accessions from seed bank collections. Here, we studied the QA profiles of these two species in 31 wild populations. We then compared the QA compositions of the wild populations with samples of the four cultivated lupin species. For this, we developed a new LC–MS/MS method for determining QAs in lupin seeds. The method was validated in-house using 15 QAs available from commercial sources. In addition, we used GC–MS to identify some QAs for which no analytical standard was available. We analyzed the inner statistical correlations between the various QAs identified in the different species and their connection with the precursor amino acid lysine to gain new insights into QA phyto-production.

## 2. Results

### 2.1. Analytical Method

A novel LC–MS/MS method was developed and validated in-house using 15 QA standards (see Appendix A for information on vendors). A straightforward one-step sample extraction procedure using acidified methanol/water was applied, leading to recoveries between 80 and 105% for all QAs except 13-trans-cinnamoyloxylupine, which displayed yields between 45 and 55% (Appendix A). In-house validation showed relative standard deviations (*n* = 6) between 2 and 14% for all compounds at the three spiking levels tested (1–5–25 mg/kg). However, QA levels in lupin seeds can be very high, exceeding 10,000 mg/kg for individual QAs in some bitter varieties, far above the level that can be realistically spiked in a sample. To check recovery at such high concentrations, several samples were extracted twice with the extraction solvent. The presence of only small amounts of QAs in the second extract (<10%) confirmed the high extraction efficiencies obtained with the spiked samples. Calibration lines showed good to excellent linearity (>0.995) over the full calibration range of 0–200 µg/L for all QAs (Appendix A). Due to the huge range of QA concentrations in the lupin seeds, sample extracts were prepared with overall standard dilution factors (DFs) of 2000, 10,000, and 50,000. This approach ensured that all QAs could be quantified against the same set of calibration lines by selecting the best-fitting diluted extract for each QA.

The available standards covered the majority of QAs present in the lupin species. Nevertheless, some relevant QA commercial standards were not available, including 11,12-seco-12,13-didehydromultiflorine and 13-hydroxymultiflorine, and several esters of 13-hydroxymultiflorine and 13-hydroxylupanine. These QAs were semi-quantified by taking a structurally related QA with high structural similarity as a reference. Identification of the QAs without reference compounds is described in the ‘Materials and Methods’ section below.

### 2.2. Quinolizidine Alkaloid Profiles

Total QA compositions by LC–MS/MS are detailed in the Appendix A. The QA amounts reported here are averages for each of the 31 tested wild *L. pilosus* and *L. palaestinus* populations. The total QA composition of four cultivated lupin species—*L. angustifolius*, *L. luteus*, *L. mutabilis*, and two cultivars of *L. albus*, as well as two control groups—chickpea and soybeans—are also reported. This research uses these species as a reference group for the wild lupins (*L. pilosus* and *L. palaestinus*), as they are more thoroughly studied. Twenty-one quinolizidine alkaloids are identified and (semi)-quantified in the samples. In Appendix A, representative LC–MS/MS chromatograms are provided for the six lupin species in this study.

Kruskal–Wallis tests on the total QA amounts reveal the uniqueness of each species tested and the apparent differences between the analyzed varieties or species (Figure 1). All populations relating to the same species show a close statistical correlation in the total amount and the profile of alkaloids produced. *L. pilosus* and *L. palaestinus* show a lower total QA content than the seeds of the other tested species, except for the *L. albus* sweet variety (Figure 1).

The heat map of all the QAs detected (Figure 2A) shows a very high inner correlation among species. Among *L. pilosus* and *L. palaestinus*, all populations relating to the same species showed a close inner correlation in QA compositions, although grown in different ecological systems (Figure 2). The two lupin species show a much closer statistical correlation to one another than to the other species, which produces a different array of QAs (Figure 2B). Regarding individual compounds, the dominant QAs in *L. pilosus* and *L. palaestinus* are multiflorine and epilupinine, the latter being less abundant in *L. palaestinus*. Other important marker QAs for both species are 11,12-seco-12,13-didehydromultiflorine and 13α-hydroxymultiflorine. Together, these four QAs comprise 95–97% of the QA composition in the two species. In the other species, the contribution of these four QAs to the total content is between 0.2% (*L. mutabilis*) and 10–11% (*L. albus*).

In *L. pilosus* and *L. palaestinus,* sparteine is present at moderate levels (Figure 3). This is also the situation for other lupin species besides *L. mutabilis*, in which sparteine is one of the marker QAs (Figure 3). Other marker QAs in *L. mutabilis* are lupanine, 13α-hydroxylupanine, and 3β,13α-dihydroxylupanine. Together, these four QAs are responsible for approx. 88% of the QA content in *L. mutabilis*.

Lupanine and 13α-hydroxylupanine are major compounds in *L. albus* and *L. angustifolius* (Figure 3). In *L. albus*, they are complemented with albine and 13α-hydroxymultiflorine (accounting for 80–87% in total), and in *L. angustifolius* with angustifoline (representing 96% in total). Finally, *L. luteus* displays a QA profile different from all the other lupin species, the marker QAs being lupinine, gramine, and, present in smaller amounts, epilupinine, which together make up 99% of the QA content. The results show that the seeds of the six species studied have distinctively different QA profiles with different QAs dominating. *L. pilosus* and *L. palaestinus* are closely related as they contain the same QAs. Nevertheless, they can be differentiated based on the ratio of epilupinine to multiflorine in the seeds, which for *L. pilosus* and *L. palaestinus* contain 44 ± 12% and 12 ± 8.5%, respectively.

Another heat map reveals the significance of the statistical correlation shown in Figure 3, pointing out the correlation between the different species among themselves and compared with other analyzed lupin species.

### 2.3. Correlation with L-Lysine

Correlation plots between l-lysine amounts (*w*/*w*% of DW) and the amounts of four main alkaloid groups present in *L. pilosus* and *L. palaestinus*—(epi)lupinine, lupanine, multiflorine, and sparteine, were constructed to reveal the connection between l-lysine levels and alkaloid production (Figure 4). (Epi)lupinine and multiflorine correlate negatively with l-lysine levels, suggesting that when lupinine and multiflorine are produced, l-lysine is consumed and its concentration drops. No significant correlation was demonstrated between l-lysine and lupanine or l-lysine and sparteine. As l-lysine is the precursor of all QA derivatives, including lupanine and sparteine, the lack of correlation may be attributed to their very low levels, suggesting that the two are under-produced in these lupin species.

## 3. Discussion

The new LC–MS/MS method developed and presented here proved effective for the detection, identification—and, most importantly—accurate quantification of QAs. Whenever a QA reference standard was unavailable, GC–MS complemented the identification. Thus, we were able to identify 21 QAs in six different species. Ten of these 21 QAs could be quantified employing a reference standard, while the others were semi-quantified based on structurally related QAs. When previously analyzed by GC–MS as a stand-alone method, relative quantification was achievable for these species [25,26,27,28]. GC–MS is often the gold standard for QA analysis in lupin seeds. This is partly because GC-based methods have been used for over 40 years, and extensive databases that tabulate the various QAs in *Lupinus* and related genera have been compiled. Only recently have the first LC–MS analytical methods started to appear [11,12,29]. The limitation of many studies, particularly GC–MS-based methods, is that primarily semi-quantitative results are reported due to the lack of suitable reference standards. Lupanine is often used as a reference standard for quantitation. Still, it should be acknowledged that the detector response is strongly dependent on the specific chemical structure of a compound as well as the measurement conditions. This variability in detector response is also an essential factor in LC–MS/MS analysis, as can be seen for the QAs incorporated in the method (Appendix A). For the QA calibration lines, the slope differences amounted to a factor of 12. Nevertheless, a large part of the QAs could be quantified using this method, resulting in a more detailed assessment of the QA content in lupin seeds.

The QA profiles obtained for the six lupin species generally agreed with data reported in the literature [15,25,26,27,28,30]. However, some differences were observed. Swiecicki et al. [25] reported that their accessions of *L. luteus* contained primarily lupinine and sparteine with smaller amounts of gramine. Similar results were reported by Wink et al. [15]. Magelhaes et al. [31], on the other hand, reported the absence of lupinine in several *L. luteus* cultivars. In our sample, sparteine was only present at trace levels. This suggests that a wider variability within *L. luteus* may exist regarding the production of marker QAs.

Interestingly, it was found that *L. pilosus* and *L. palaestinus* produce epilupinine rather than lupinine. The literature so far provides contradicting results. For instance, Wink et al. [15] and Święcicki et al. [25] reported lupinine as the predominant epimer. However, Ainouche et al. [32] reported epilupinine as the epimer present in both species. Our study confirms the original findings of Thomas et al. [33], who specifically reported epilupinine and not lupinine as present in *L. pilosus*. Epilupinine and lupinine have close retention times in most GC–MS methods, which may have led to misinterpretation of the data by some authors.

The average total QA content determined in the studied *L. pilosus* and *L. palaestinus* populations was 10,250 ± 1000 and 6040 ± 1090 mg/kg, respectively. Swiecicki et al. [25] reported a range from 872 to 4717 mg/kg for *L. pilosus* (six accessions) and a range from 774 to 2308 mg/kg for *L. palaestinus* (six accessions). The statistical differences between the accessions were significant (*p* < 0.01), which was attributed by Swiecicki et al. [25] to different gene bank origins of the samples. The levels reported by Swiecicki et al. [25] are substantially lower than those determined in our current study, and differences between accessions were also not statistically significant. Swiecicki et al. [25] used lupanine as the single standard for quantification of the different QAs detected in the seeds, and it cannot be excluded that the QA content was underestimated due to differences in detection response between the QAs present in *L. pilosus* and *L. palaestinus*, and lupanine.

Some publications regarding alkaloid biosynthesis in plants, including genera closely related to *Lupinus*, suggested some QA biosynthesis pathways in lupins [7,16]. These suggested QA biosynthesis pathways in lupins were extrapolated from known pathways of different alkaloids and their derivative compounds produced by other (non-lupin) species. Our results present a new comprehensive dataset derived from QAs in lupin species, which directly supports our refined suggested pathway (Figure 5).

Statistical data analysis allows us to revisit the QA biosynthesis mechanisms in lupin seeds. As mentioned above, the QA compositions and QA correlations within species observed here confirm the QA formation biosynthetic pathway (Figure 5). By integrating all the accumulated data, we suggest that the previously suggested biosynthesis pathways of QAs in lupins are a targeted path. The statistical analysis of individual QAs at the species level allows us to explore further nuances in lupin QA biosynthesis. Plots of the QA compositions produced by the different species show that any given species produces a set of QAs and refrains from producing other QAs (Figure 3, Appendix A). For example, *L. pilosus*, as observed in the various populations sampled, produces relatively high amounts of multiflorine but almost no lupanine or related derivatives, including angustifoline. On the other hand, *L. albus* seeds contain relatively high amounts of lupanine but hardly any (epi)lupinine (Figure 3, Appendix A).

This preferential production of certain assemblages of QAs is consistent in all species studied. We show that (1) only certain QAs will be produced for any variety of lupin, and others will not, and (2) there is a link between the main QAs that will be produced by a certain lupin species (Figure 4). From this, we can extrapolate that the synthases governing QA production are not randomly activated but are rather controlled by a ‘switch on/switch off’ system. We suggest that lupins produce specific dual QA combinations, i.e., they switch on only two out of the four main QA pathways at any given time. These combinations may be either (epi)lupinine coupled with multiflorine (as in *L. pilosus* and *L. palaestinus*), multiflorine and lupanine (as in *L. albus* and *L. angustifolius*), sparteine and lupanine (as in *L. mutabilis*), or lupinine and sparteine (as in *L. luteus*). Thus, even though the common precursor l-lysine can produce all QAs detected in lupins, is seems that when lupanine and its derivatives are produced, only a determined dual combination will result for each variety. Similar observations, leading to similar conclusions, were independently drawn by Swiecicki et al. [25], although their analysis was based on a smaller number of lupin populations and fewer QAs.

The previously suggested pathway for QA production in lupins identified the amino acid l-lysine as the precursor of most QAs studied. Here, we propose a new way to examine the QA biosynthesis pathway. Our results support the concept of l-lysine as the amino acid precursor for QA biosynthesis in lupin seeds, as demonstrated in Figure 5. For epilupinine and multiflorine, which are the dominant alkaloids in both *L. pilosus* and *L. palaestinus*, the lower the l-lysine amount detected in the seeds, the higher the total alkaloids accumulated (Figure 4A,B). This inverse correlation reinforces the suggestion that l-lysine is the precursor for alkaloids produced by lupins. Notably, sparteine and lupanine, produced in negligible or low amounts in the two wild species, showed no correlation with l-lysine concentrations (Figure 4C,D). This accords with the recognition of l-lysine as the precursor for QA biosynthesis and further supports a preferential or alternate production when one production flow is dominantly executed over the other, as described in Figure 4. In *L. pilosus* and *L. palaestinus*, for example, it appears that l-lysine is preferably transformed into multiflorine and epilupinine, while the production of lupanine is very low. Lupinine derivatives were not detected in these species when screened for using GC–MS. In contrast, 13α-tigloyloxymultiflorine and 13α-tigloyloxylupanine, two prominent multiflorine and lupanine ester derivatives, were detected in the lupin seed extracts (Figure 5).

## 4. Conclusions

This paper is an essential step in describing the QA profiles of local wild lupin varieties to be used as the basis to transform them into agricultural crops, either for human consumption or for stock animal feed. We can gain some insights from the cultivation process of *Lupinus angustifolius* in the 1930s. For many centuries, until the appearance of its low-alkaloid mutants, *L. angustifolius* was used mainly as a green manure because the high QA content precluded its use as a feed ingredient. This is a bi-facial phenomenon as low-alkaloid (widely named ‘sweet’) genotypes are mandatory for edible crops but make the plants more susceptible to pathogens and diseases. Some researchers [34] suggested creating ‘bitter/sweet’ dual varieties, where some varieties of the same species will be high in QA content (bitter) and can be used for green manure, while low-QA varieties will be used for food and feed. Creating varieties with high levels of QAs in the vegetative organs while retaining low QA concentrations in edible seeds [34] may lead to a dual-purpose plant that preserves its adaptive fungicidal, antibacterial, and insecticidal capabilities.

Another important motivation for this effort is the desired biological activity that QAs have demonstrated in multiple studies [35]. Many QAs show beneficial and desired qualities such as antioxidant, antimicrobial, anti-carcinogenic, and anti-inflammatory activities [23,36,37,38,39,40,41,42,43,44,45,46,47]. However, several individual alkaloids, especially sparteine, anagyrine, and cysteine, have undesired or even toxic activity, thus their concentrations in edible lupins must be strongly regulated [4,47]. However, cysteine and anagyrine concentrations in the wild lupin species studied here were below the detection limit of the LC–MS/MS (0.2 mg/kg). Thus, we conclude that the wild populations of native lupins in Israel might have a future as promising legume crops.

## 5. Materials and Methods

### 5.1. Materials

Seven lupin species were included in this study. Mature pods of two wild populations of (A) *L. pilosus* and (B) *L. palaestinus* were sourced in Israel during late spring (see Section 5.2.1 below) (Table 1). Twenty populations of *L. pilosus* and eleven populations of *L. palaestinus* were sampled. Additionally, seven different species or varieties of lupins were included as a reference: four different sources of *L. albus* (two bitter and two sweet), *L. angustifolius*, *L. luteus*, and *L. mutabilis*. Two non-lupin plants—chickpea (*Cicer arietinum*) and soybean (*Glycine max*)—were included as controls, as they do not contain QAs. The low levels of QAs in these control samples are most likely due to minor contamination during sample preparation/analysis. Five repetitions (containing different seeds each time) per each of the 40 samples were collected to compose an entire set of 200 seed samples. The sampling unit was a fixed number of seeds adjusted to each species and cultivation.

### 5.2. Methods

#### 5.2.1. Seed Collection Method

Wild populations of *L. pilosus* and *L. palaestinus* were surveyed in natural areas in Israel, representing a range of environmental conditions (including various climate and soil conditions) typical of Eastern Mediterranean ecosystems. Populations were identified and marked in early spring (February–early March) at the beginning of the flowering period. In late spring (April–May), matured lupin pods were collected before seed dispersal occurred. Around 100 individuals were sampled at each site. Matured pods standing on plants were collected into paper bags and brought to Tel Aviv University laboratories. Once in the lab, pods from each population were placed in separate containers and left to dry at room temperature until all pods opened naturally. Dried lupin pods were removed, and seeds from each population were kept in dry and cool conditions.

#### 5.2.2. Analytical Standards

Fifteen quinolizidine alkaloids were available as analytical standards for this study: albine, anagyrine, angustifoline, 13α-*trans*-cinnamoyloxylupanine, cytisine, epilupinine, gramine, 13α-hydroxylupanine, isolupanine, lupanine, lupinine, methylcytisine, multiflorine, sparteine, and thermopsine. See Appendix A for further details on vendors and purity. Individual QA stock solutions of 200 µg/mL were prepared in methanol. Mixed QA solutions in methanol (5 µg/mL, 500 ng/mL) were prepared from the individual stocks. LC–MS-grade methanol was obtained from Actu-All (Oss, The Netherlands). Analytical-grade formic acid and ammonium carbonate were purchased from Sigma-Aldrich (Zwijndrecht, The Netherlands) for HPLC analyses. Ethyl acetate and ammonia (NH_4_OH, 25%) were HPLC grade. Deionized water (Milli-Q, Merck Millipore, Darmstadt, Germany) with a minimum resistance of 18.2 MΩ was used.

#### 5.2.3. Sample Pretreatment

Samples were ground to a particle size of <1 mm using a FOSS grinder (FOSS CM-290 Cemotec, Labtec, Auckland, New Zealand). Five replicates of 2 g were weighed into 50 mL polypropylene tubes with screw caps, and 40 mL of methanol/water (1:1 *v*/*v*) containing 1% formic acid was added. The mixture was shaken vigorously for 30 min on a rotary tumbler and then centrifuged for 15 min at 3500 rpm. A total of 0.5 mL of the supernatant was transferred and diluted with water up to 50 mL, and 500 µL of the diluted sample extract was transferred to a 500 µL vial. Extracts containing QAs exceeding the range of the calibration line (see below) were diluted 5- and 25-fold.

#### 5.2.4. Liquid Chromatography–Tandem Mass Spectrometry (LC–MS/MS)

Analysis was performed with an LC–MS/MS system consisting of a Waters Acquity UPLC coupled to a Xevo TQ-S tandem mass spectrometer (Waters, Milford, MA, USA). The alkaloids were separated on an Acquity UPLC BEH C18, 1.7 µm, 100 × 2.1 mm column (Waters, Milford, MA, USA). The solvents used were (A) ammonium carbonate buffer (10 mmol/L, pH 9.0 ± 0.1) and (B) methanol. At a flow rate of 0.4 mL/min, the linear gradient conditions were 0.0–1.0 min, 0% B; 8.0 min, 40% B; 12.0 min, 80% B; 12.2 min, 0% B; and 12.2–14.2 min, 0% B. The injection volume was 2 µL, and the column oven temperature was 50 °C. A solvent discard was included at 0–1.5 min and 13–14.2 min.

The instrument was run in the positive electrospray ionization (ESI) mode. The capillary voltage used was 3.0 kV with 30 V cone voltage. The source temperature was set at 150 °C, desolvation temperature at 600 °C, cone gas flow at 150 L/h, and desolvation gas flow at 800 L/h. Argon was used as the CID gas at a pressure of 10^−3^ mbar (0.17 mL/min). Two to three multiple reaction monitoring (MRM) transitions were measured per analyte. See Table 2 for detailed information on MRM transitions, MS settings, and retention times. Masslynx 4.2 and TargetLynx 4.1 software (Waters, Milford, MA, USA) were used for data acquisition and processing, respectively.

Lupin QAs for which an analytical standard was available were quantified by external matrix calibration using a diluted blank soybean extract. A nine-point calibration line (0–200 ng/mL) was constructed for each analyte and injected before and after each set of samples. The linearity of the calibration lines was >0.999. The recovery was monitored by spiking a blank soybean sample with a mixture of the QA standards (5 µg/mL) corresponding to a level of 10 mg/kg. The amount retrieved after extraction was used to correct the results for recovery. The LOQ of the method was defined as the lowest concentration validated (1 mg/kg). The LOD was not determined in detail but was visually estimated to be between 0.1 and 0.5 mg/kg, depending on the QA. Due to the wide range of concentrations found for individual QAs in the seeds (1 → 10,000 mg/kg), the sample extracts were measured undiluted, as well as 5- and 25-fold dilutions. This corresponds to concentration ranges of 0–400 mg/kg, 0–2000 mg/kg, and 0–10,000 mg/kg, respectively.

QAs for which no analytical standard was available were semi-quantified by comparison with a standard of similar structure, as indicated in Table 2.

Each set of 50 extracts included 4 lupin extracts with known content as quality control (QC) samples. These QC samples were used to check the reproducibility of the measurements. Reproducibility for individual compounds varied between 2 and 16%, and for the total PA content, between 4 and 8%, showing good LC–MS/MS system stability and reproducibility between the measurements.

#### 5.2.5. Validation of the LC–MS/MS Method

The method was validated in-house by spiking finely ground samples (0.5 g) at three levels (1, 5, 25 mg/kg, six replicates each) of a mixture of QAs. Because no blank lupin seeds were available, soybeans were used as the surrogate blank material. Soybeans were chosen as they have a composition comparable to lupin seeds but do not contain QAs. The results for recovery and repeatability of the method are presented in Appendix A. The linearity of the combined calibration lines (run before and after the samples) is shown in Appendix A.

#### 5.2.6. Gas Chromatography Coupled with Mass Spectroscopy (GC–MS)

Sample preparation, GC–MS analysis, and QA identification followed established protocols [23,48,49,50], specifically the procedure entitled A2 by the authors in [50]. In summary, the main steps are described below.

Lupin samples (15 g from each lupin species included in the project, see Table 1) were freeze-dried for 48 h (Freeze dryer LMC-2, Martin Christ, Osterode, Germany) and then ground to a fine powder using a FOSS grinder (FOSS CM-290 Cemotec, FOSS Ltd., Nils Foss Allé 1, DK-3400 Hilleroed, Denmark). The QA extraction method was developed in-house following the procedure described by Kushnareva et al. [50]. A total of 2 g of ground lupin seeds were weighed into a test tube, and 9 mL of a solvent mixture consisting of ethyl acetate and ammonia solution (NH_4_OH, 25%), in an 8:1 volumetric ratio, was added to the ground lupin samples and left for 18 h in a refrigerator (4 °C). The extracts were filtered twice, once through a 0.45 µm membrane filter paper, followed by a 0.22 µm membrane filter, and directly injected into the GC–MS equipment.

GC–MS analyses were carried out using an Agilent 7890B gas chromatograph coupled to a 5977A mass spectrometer (electron multiplier potential 2 kV, filament current 0.35 mA, electron energy 70 eV), and the spectra were recorded over the range of *m*/*z* 40 to 500. An Agilent 7683 autosampler was used for the sample introduction. A 1 μL aliquot of each sample was injected into the GC–MS equipment using a 1:10 split-ratio injection mode. Helium was the carrier gas at a constant flow of 1.1 mL/s. An isothermal hold at 50 °C was maintained for 2 min, followed by a heating gradient of 6 °C/min to 300 °C, and the final temperature was held for 4 min. A 3-minute solvent delay was applied. A 30 m × 0.25 mm ID, 5% cross-linked phenylmethyl siloxane capillary column (HP-5MS) with a 0.25 μm coating thickness was used for separation, and the injection port temperature was 220 °C. The MS interface temperature was 280 °C.

Peaks were assigned with a spectral library (NIST 17.0) and compared with MS data obtained from injecting 12 analytical standards. For identification and relative quantification, 10 µL of the QA standard mixtures (5 µg/mL, corresponding to a level of 10 mg/kg) were injected into the GC–MS equipment, and their retention time and elution order were set. Identification of QAs present in LC–MS/MS chromatograms, for which no correlating standards were available for confirmation, was made based on cross-referencing with 2 NIST libraries (NIST2008, NIST2014) and verified for molecular ion mass and typical six dominant breakdown ions using the library search engine. In this way, the identity could be confirmed of 12 QAs in the LC–MS/MS measurements.

Ten lupin seeds for each of the lupin varieties included in the project (see Table 1) were freeze-dried for 24 h (Freeze dryer LMC-2, Martin Christ, Osterode, Germany), then milled to a fine powder using a FOSS grinder (FOSS CM-290 Cemotec, FOSS Ltd., Nils Foss Allé 1, DK-3400 Hilleroed, Denmark). A total of 50 mg of dried powder was added to 300 mL of distilled water to allow the extraction of the proteins. Protein hydrolysis was carried out using ETHOS EASY Advanced Microwave system with an HPR-3000/1 rotor [50]. Lysine content was determined by using 2,4-dinitrofluorobenzene (DNFB) for derivatization. The amino-acid-derived samples were separated on a reversed-phase C18 column (Luna^®^ Omega 3 µm, 100 Å; 150 × 4.6 mm) using an Ultimate 3000 UHPLC (Thermo Fisher Scientific, 168 Third Avenue, Waltham, MA, USA, 02451) following the method by [51].

The boxplot comparison analysis and hierarchical clustering were performed using R software (version 4.0.2, Team 2020). To compare the total QA variable, the assumptions of normality and homogeneity of residual variance were first assessed by visually plotting sample quantiles against theoretical quantiles and residuals against fitted values, followed by performing Shapiro–Wilk and Levene’s tests. The variable for total QAs did not meet the above assumptions, even after applying square root and log transformations and, therefore, was treated as nonparametric. Due to the nonparametric nature of the data, a Kruskal–Wallis test followed by a Dunn post-hoc test was used to assess differences between species using the *kruskal.test* base function and *dunn.test* function of the *FSA* package [52]. Due to the absence of QAs in the *L. albus* sweet samples, these were removed from the analysis presented in Figure 3 to avoid statistical bias and, therefore, marked with an asterisk sign (*) instead of letters as with the other tested species.

The hierarchical clustering and production of heat maps for the different lupin species and QAs were carried out using the as.dendrogram function of the *pheatmap* package [53] after being scaled using the *scale* base function.

## Figures and Tables

**Figure 1 toxins-16-00163-f001:**
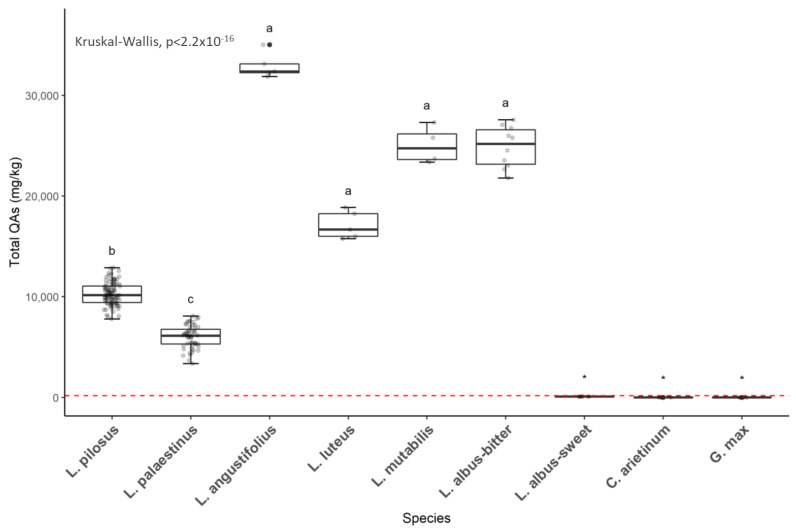
Total quinolizidine alkaloid (QA) levels were detected in wild local species *L. palaestinus* and *L. pilosus* (in the grey box) compared with other lupin or legume species. The Australian regulatory threshold for human consumption (200 mg QAs/kg seed) is marked with a dashed red line. Kruskal–Wallis significance (*p* > 0.05) is marked with small letters. Groups marked with an asterisk were used as control groups and thus were not included in the statistical analysis.

**Figure 2 toxins-16-00163-f002:**
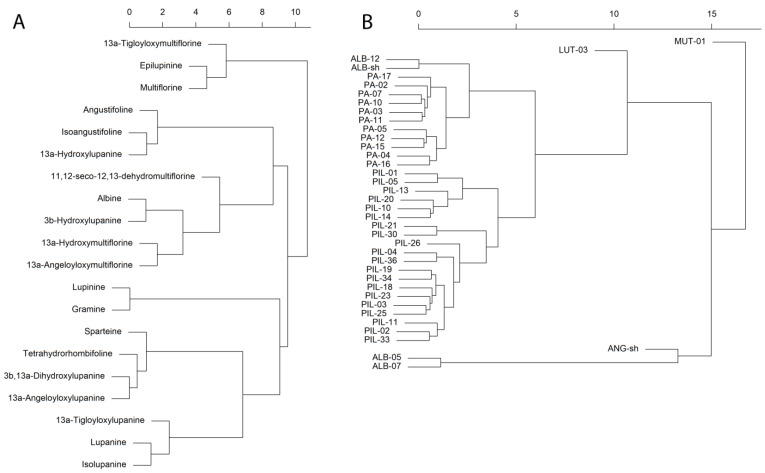
Hierarchical clustering of the studied lupin species and their QAs. The hierarchical clustering (**A**) shows the statistical distances between the various QAs identified. The grouping of the different lupin populations (**B**) is based on their complete QA profiles, as correlated in (**A**). Analysis was carried out using the dendrogram function of the *pheatmap* package after scaling using the scale base function.

**Figure 3 toxins-16-00163-f003:**
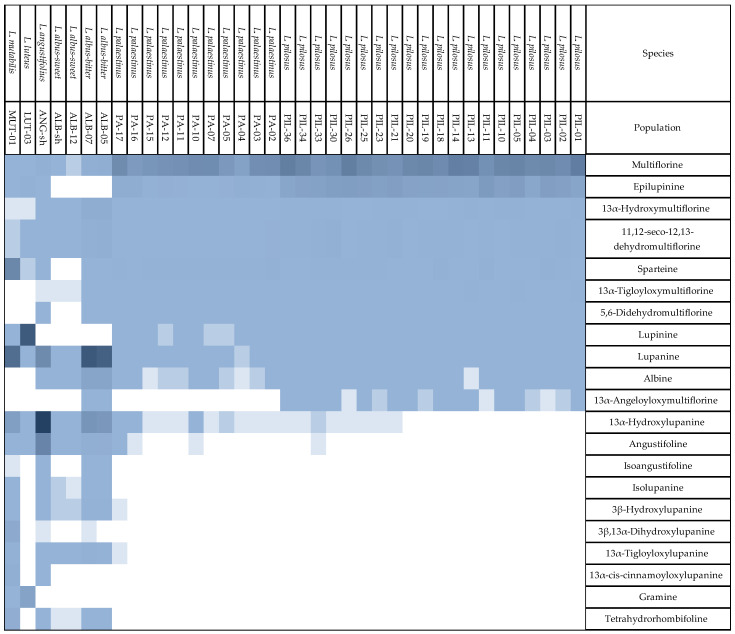
Heat map of the relative ranked quantity of the individual QAs in the analyzed lupin populations and control groups. Results are shown in a grayscale relative gradient, from the highest concentration (in blue) to the lowest/non-detected concentrations (in white). For complete details see Appendix A.

**Figure 4 toxins-16-00163-f004:**
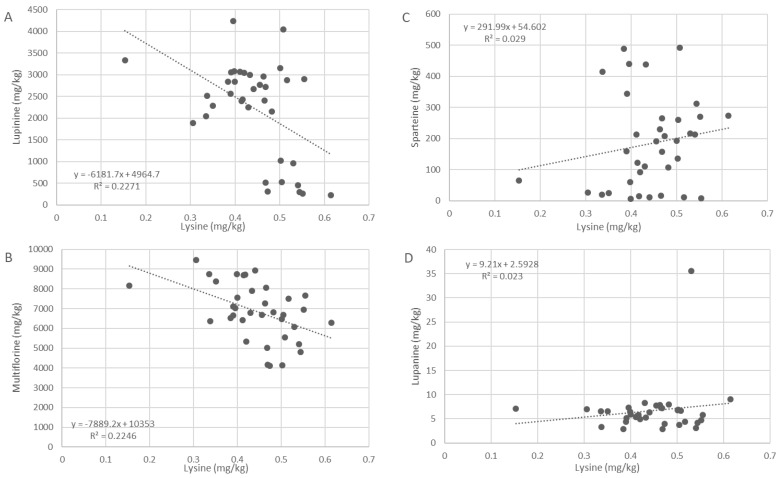
Correlation plots between l-lysine amounts (*w*/*w*% of DW) and the amounts of the four main alkaloid groups—(**A**) (epi)lupinine, (**B**) multiflorine, (**C**) sparteine, and (**D**) lupanine, in *L. pilosus* and *L. palaestinus*.

**Figure 5 toxins-16-00163-f005:**
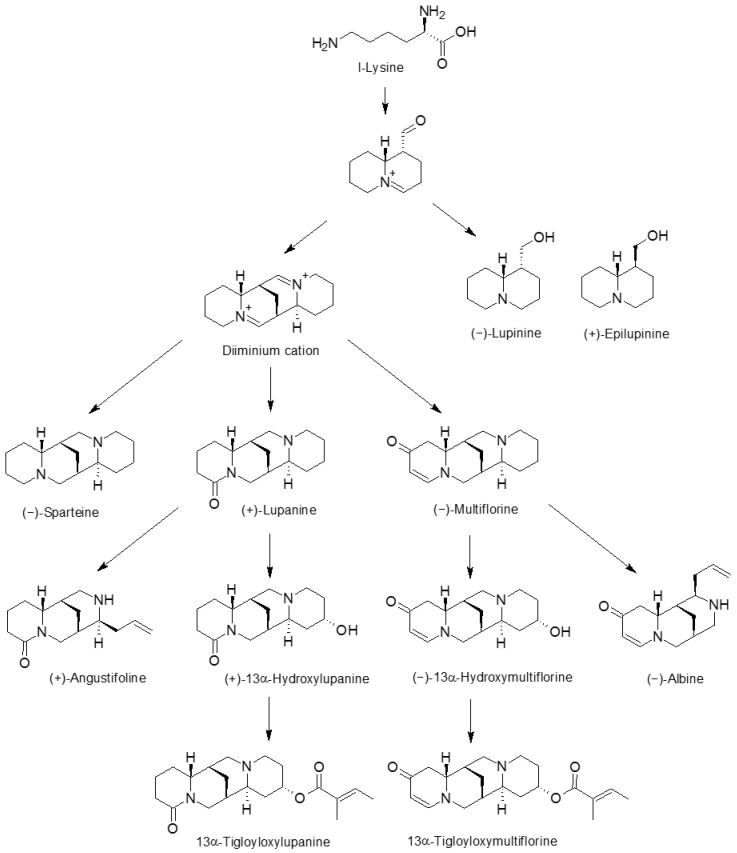
Proposed biosynthesis pathway of quinolizidine alkaloids in lupin seeds based on the observations described in the text.

**Table 1 toxins-16-00163-t001:** List of the samples included in this study.

Species	Regional Distribution	Population Code	Collection Site/Notes
	Golan Heights	PIL-1	Mapalim Junction
	Hula Valley	PIL-2	Nahal Hamdal
	Judean Hills	PIL-3	Tel Socho South
	Golan Heights	PIL-4	Ofir Viewpoint
	Golan Heights	PIL-5	South-west Hispin
	Golan Heights	PIL-10	Hazeka Road
	Judean Hills	PIL-11	Matta
	Golan Heights	PIL-13	Fares Road (vineyard)
	Golan Heights	PIL-14	Tel Fazra
*Lupinus pilosus*	Judean Hills	PIL-18	Khirbet Kanim
	Judean Hills	PIL-19	Zechariya (Tel e-Sharia)
	Golan Heights	PIL-20	Avital
	Carmel Mountain	PIL-21	Makura
	Samarea Mountains	PIL-23	Awartha
	Samarea Mountains	PL-25	Shechem Mountains
	Samarea Mountains	PIL-26	Kedumim
	Carmel Mountain	PIL-30	Kerem Maharal South
	Judean Hills	PIL-33	Sarisa
	Lower Galilee	PIL-34	Ahuzat Barak
	Lower Galilee	PIL-36	Nau’ra
*Lupinus palaestinus*	Sharon—Coastal Plain	PA-2	Pardes Hana
	Sharon—Coastal Plain	PA-3	Ilanot Forest West
	Coastal Plain (East)	PA-4	Yashresh
	Sharon—Coastal Plain	PA-5	Bnei-Tzion North
	Sharon—Coastal Plain	PA-7	Ilanot Forest East
	Sharon—Coastal Plain	PA-10	Netanya
	Sharon—Coastal Plain	PA-11	Hirbet Samara
	Sharon—Coastal Plain	PA-12	Hod Ha-Sharon
	Sharon—Coastal Plain	PA-15	Tel Mond—Kurkar
	Coastal Plain (South)	PA-16	Ashqelon
	Coastal Plain (East)	PA-17	Sitriya
*Lupinus albus* (bitter)	Egypt, Central Nile	ALB-05 *	
	Egypt, Central Nile	ALB-07 *	
*Lupinus albus* (sweet)	Commercial Product 1	ALB-12	
	Commercial Product 2	ALB-sh	
*Lupinus angustifolius*	Sharon—Coastal Plain	ANG-sh *	Haogen
*Lupinus luteus*		LUT-03	
*Lupinus mutabilis*	The Netherlands	MUT-01 **	*Lupinus mutabilis*, var. cruickshankii. Vreeken’s Zaden Seed Company, Dordrecht, The Netherlands, www.vreeken.nl (accessed on 12 March 2024)
*Glycine max* (soybean)		MAX-01	Organic soybeans certified by Pro-Cert 2020 Crop year, Thompson Limited Canada, Net—25 kg, HOSSQ21-24, Packed 03/2021, Sell by 03/2022
*Cicer arietinum* (chickpea)		ARI-01	Organic Chickpeas by Tvuot, www.tvuot.co.il (accessed on 12 March 2024), 500 g package Best before 01/06/2022

* Non-edible variety (high QA levels). ** Non-edible species (used as an ornamental garden species).

**Table 2 toxins-16-00163-t002:** MS/MS MRM transitions.

No.	Compound	Precursor Ion (*m*/*z*)	Product Ion 1 (*m*/*z*)	Col. Energy 1 (eV)	Product Ion 2 (*m*/*z*)	Col. Energy 2 (eV)	Product Ion 3 (*m*/*z*)	Col. Energy 3 (eV)	Indicative RT (min)	MRM	Compound Quantified
	Standard QAs										
1	Gramine	130.0	77.0	20	103.0	25			7.35	1	1
2	Epilupinine	170.2	96.0	30	152.0	20	124.0	25	5.50	1	2
3	Lupinine	170.2	124.0	25	152.0	20	96.0	30	6.00	1	3
4	Cytisine *	191.2	133.0	30	148.0	20			5.70	1	4
5	Methylcytisine *	205.2	58.2	20	108.0	20			7.20	1	5
6	Albine	233.2	112.0	20	138.0	20	150.0	30	7.15	2	6
7	Angustifoline	235.2	112.0	20	193.0	30	114.0	30	8.15	2	7
8	Sparteine	235.2	98.0	30	233.0	30	70.0	30	7.65	2	8
9	Anagyrine *	245.2	70.0	45	98.0	35			9.40	2	9
10	Thermopsine *	245.2	70.0	45	98.0	35			9.50	2	10
11	Multiflorine	247.2	70.0	40	112.0	25	134.0	20	7.65	2	11
12	Lupanine	249.2	114.0	30	136.0	30	98.0	30	8.10	3	12
13	Isolupanine	249.2	84.0	30	98.0	30	136.0	30	10.50	3	13
14	13α-Hydroxylupanine	265.2	114.0	30	152.0	30	84.0	40	6.95	3	14
15	13α-*trans*-Cinnamoyloxylupanine *	395.2	112.0	30	247.0	30	98.0	40	11.55	1	15
	Non-standard QAs										
16	Isoangustifoline	235.2	112.0	20	193.0	30	114.0	30	8.00	2	7
17	5,6-Didehydromultiflorine	245.2	70.0	45	98.0	35			7.60	2	9
18	11,12-seco-12,13-Didehydromultiflorine	247.2	98.0	30	112.0	25	70	40	9.95	2	11
19	Tetrahydrorhombifoline	249.2	114.0	30	136.0	30	166.0	30	11.40	3	13
20	13α-Hydroxymultiflorine	263.2	112.0	30	245.0	20	70.0	40	5.65	3	14
21	3β-Hydroxylupanine	265.2	114.0	30	152.0	30	112.0	35	6.80	3	14
22	3β,13α-Dihydroxylupanine	281.2	130.0	25	152.0	25	101.0	35	5.95	3	14
23	13α-Tigloyloxymultiflorine	345.2	112.0	30	245.0	20	70.0	40	10.10	1	15
24	13α-Angeloyloxymultiflorine	345.2	112.0	30	245.0	20	70.0	40	10.20	1	15
25	13α-Tigloyloxylupanine	347.2	112.0	30	247.0	30	98.0	40	10.70	1	15
26	13α-Angeloyloxylupanine	347.2	112.0	30	247.0	30	98.0	40	10.85	1	15
27	13α*-cis*-Cinnamoyloxylupanine	395.2	112.0	30	247.0	30	98.0	40	11.15	1	15

* Standard is included in the method but not detected in any of the samples.

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
