# Peer review of "New Analytical Approach to Quinolizidine Alkaloids and Their Assumed Biosynthesis Pathways in Lupin Seeds"

_toxins, 2024, doi:10.3390/toxins16030163_

Round 1

Reviewer 1 Report

Comments and Suggestions for Authors

The authors base their study on the arguments that bitterness in lupine species due to QAs is not a favorable trait and thus the technique they have devised could be used for screening for safe human consumption. They have not actually discussed the safe limits of the alkaloids for humans or animals. They have themselves given reasons that crop improvements leading to allocation of resource i.e QAs into vegetative part would be better since it would give the plant innate capacity to fight the pathogenesis or herbivory. Besides, when we are dealing with wild species screening for low or null QAs in seeds is a good strategy; however, if they retain this characteristic in subsequent generations would be worthwhile to study. Moreover, the accumulation of SMs has often been correlated with the environmental conditions.

The study has been designed and performed adequately and the publication has good quality and thus could be accepted.

I have few suggestions given below…

The quality of the figures is good but could be given in color….

[21] in evaluating

[63]in vacuoles of plant organs

[69]([4] and references therein)??

[69]-[71] Rewrite the sentence

[91]-[94] Rewrite the sentence

[182]-[184] Rewrite the sentence

[341] can be used

[398] 18.2 mΩ.cm or 18.2 MΩ instead of 18.2 M

Author Response

We thank reviewer 1 for their thoughtful reading of our manuscript. The reviewer's suggested that the introduction can be improved in providing sufficient background and include all relevant references. The following is our point-by-point response to the Reviewer's comments:

Comment 1: The authors base their study on the arguments that bitterness in lupine species due to QAs is not a favorable trait and thus the technique they have devised could be used for screening for safe human consumption. They have not actually discussed the safe limits of the alkaloids for humans or animals. They have themselves given reasons that crop improvements leading to allocation of resource i.e QAs into vegetative part would be better since it would give the plant innate capacity to fight the pathogenesis or herbivory. Besides, when we are dealing with wild species screening for low or null QAs in seeds is a good strategy; however, if they retain this characteristic in subsequent generations would be worthwhile to study. Moreover, the accumulation of SMs has often been correlated with the environmental conditions.

Response 1: Our research aims to shed light on alkaloid biosynthesis and underscores the potential of wild lupins as a novel crop. The bitterness of lupins, as an obstacle for crop management, and the molecules attributing this bitterness is the score of our further research. It was not addressed specifically here and thus was only mentioned in a nutshell.

Comment 2: The quality of the figures is good but could be given in color.

Response 2: Table 3 was converted to two-tone colored and replaced. A dashed red line was added to Figure 1.

Comment 3: [21] in evaluating

Response 3: DONE

Comment 4: [63]in vacuoles of plant organs

Response 4: DONE

Comment 5: [69]([4] and references therein)??

Response 5: We refer the reader to the references in the cited work for further examples.

Comment 6: [69]-[71] Rewrite the sentence

Response 6: DONE

Comment 7: [91]-[94] Rewrite the sentence

Response 7: DONE

Comment 8: [182]-[184] Rewrite the sentence

Response 8: DONE

Comment 9: [341] can be used

Response 9: DONE

Comment 10: [398] 18.2 mΩ.cm or 18.2 MΩ instead of 18.2 M

Response 10: DONE

Attached, please find a detailed list of our corrections. These corrections include those pointed by all reviewers, and other issues that we noticed and decided to improve independently. 

Reviewer 2 Report

Comments and Suggestions for Authors

The authors of the manuscript titled "New analytical approach to quinolizidine alkaloids and their assumed biosynthesis pathways in lupin seeds" have collected and analyzed seeds from 31 populations of L. pilosus and L. palaestinus (species native to Israel), to understand their alkaloid profiles, developing a method using LC-MS/MS and verifying with GC-MS. This research sheds light on alkaloid biosynthesis and underscores the potential of wild lupins as a novel crop. The research is well thought out and well written. I therefore suggest minor revisions.

Some errata that the authors should correct in the abstract, key contribution, introduction and some parts of the manuscript L. pilosus and L. palaestinus are not italicised.

Line 457 given A nine-point calibration line (0-200 ng/mL), but in Table S3 it says 8 points. Clarify whether it is 8 or 9.

Supplementary material: I think this title “Supplementary Figure S1A-B. Quinolizidine alkaloid structures are included in the analytical method” is not necessary. Indicate only the description of the figure at the bottom.

Author Response

We thank reviewer 2 for their thoughtful reading of our manuscript. The following is our point-by-point response to the Reviewer's comments:

Comment 1: Some errata that the authors should correct in the abstract, key contribution, introduction, and some parts of the manuscript L. pilosus and L. palaestinus are not italicised.

Response 1: DONE throughout the entire manuscript

Comment 2: Line 457 given A nine-point calibration line (0-200 ng/mL), but in Table S3 it says 8 points. Clarify whether it is 8 or 9.

Response 2: 9. Corrected in Table S3

Comment 3: Supplementary material: I think this title “Supplementary Figure S1A-B. Quinolizidine alkaloid structures are included in the analytical method” is not necessary. Indicate only the description of the figure at the bottom.

Response 3: DONE. Please see all our changes in the attached file "corrections list".

Attached, please find a detailed list of our corrections. These corrections include those pointed by all reviewers, and other issues that we noticed and decided to improve independently. 

Reviewer 3 Report

Comments and Suggestions for Authors

In the present manusript, the authors describe a highly useful method for the extraction and structural determination of quinolizidine alkaloids obtained from natural samples. The research has been executed with great care and the analytical method has been described in detail. The provided Supporting Information appears to be very useful. Overall, the manuscript has been written very well and in good English. The only improvement I  would like to recommend results from the fact that two recent review articles on quinolizidine alkaloids have not been cited and thus, a reference to both articles should be given at the appropriate positions described below:

1. On page 1, lines 39–41, the authors say: "QAs are characterized by their quinolizidine ring structure [4] and are produced as secondary metabolites by members of the genus Lupinus [5-6]." At the end of this sentence, the following comprehensive review on quinolizidine alkaloids, which is actually more recent than the two articles cited in the references [5] and [6], should be cited: Michael, J. P. In The Alkaloids, Vol. 75; Knölker, H.-J., Ed.; Academic Press: London, 2016; pp 1–498.

2. On page 2, lines 50–52, the authors say: "Despite their potential uses for medicine and plant protection, information on thetoxicity of QAs occurring in Lupinus species is limited [4, 9], and so far, no comprehensive human health risk has been established [4, 10-13]." At the end of this sentence, the following comprehensive review on the biological activities of quinolizidine alkaloids should be cited: Zhang, J.; Liu, Y.-Q.; Fang, J. In The Alkaloids, Vol. 89; Knölker, H.-J., Ed.; Academic Press: London, 2023; pp 1–37.

With the minor revision decribed above, this excellent article is strongly recommended for publication.

Author Response

We thank reviewer 3 for their thoughtful reading of our manuscript. The following is our point-by-point response to the Reviewer's 3 comments:

Comment 1: In the present manuscript, the authors describe a highly useful method for the extraction and structural determination of quinolizidine alkaloids obtained from natural samples. The research has been executed with great care and the analytical method has been described in detail. The provided Supporting Information appears to be very useful. Overall, the manuscript has been written very well and in good English. The only improvement I would like to recommend results from the fact that two recent review articles on quinolizidine alkaloids have not been cited and thus, a reference to both articles should be given at the appropriate positions described below:

  1. On page 1, lines 39–41, the authors say: "QAs are characterized by their quinolizidine ring structure [4] and are produced as secondary metabolites by members of the genus Lupinus [5-6]." At the end of this sentence, the following comprehensive review on quinolizidine alkaloids, which is actually more recent than the two articles cited in the references [5] and [6], should be cited: Michael, J. P. In The Alkaloids, Vol. 75; Knölker, H.-J., Ed.; Academic Press: London, 2016; pp 1–498.

Response 1: DONE.

  1. On page 2, lines 50–52, the authors say: "Despite their potential uses for medicine and plant protection, information on the toxicity of QAs occurring in Lupinus species is limited [4, 9], and so far, no comprehensive human health risk has been established [4, 10-13]." At the end of this sentence, the following comprehensive review on the biological activities of quinolizidine alkaloids should be cited: Zhang, J.; Liu, Y.-Q.; Fang, J. In The Alkaloids, Vol. 89; Knölker, H.-J., Ed.; Academic Press: London, 2023; pp 1–37.

Response 2: Added to the discussion section

Attached, please find a detailed list of our corrections. These corrections include those pointed by all reviewers, and other issues that we noticed and decided to improve independently. 
